behaviour

mortality risk, survival, RFID, foraging, orientation flight, behavioural experience

**Authors for correspondence:**
Alberto Prado
e-mail: aprado@unam.mx
Fabrice Requier
e-mail: fabrice.requier@egce.cnrs-gif.fr

# Honeybee lifespan: the critical role of pre-foraging stage

Alberto Prado[1,†], Fabrice Requier[2,†], Didier Crauser[3], Yves Le Conte[3], Vincent Bretagnolle[4,5] and Cédric Alaux[3]

[1]Escuela Nacional de Estudios Superiores, Unidad Juriquilla, UNAM Querétaro, Querétaro, Mexico
[2]Université Paris-Saclay, CNRS, IRD, UMR Évolution, Génomes, Comportement et Écologie, 91198 Gif-sur-Yvette, France
[3]INRAE, Abeilles and Environnement, 84914 Avignon, France
[4]Centre d'Etudes Biologiques de Chizé, CNRS and La Rochelle University, UMR 7372, 79360 Beauvoir sur Niort, France
[5]LTSER Zone Atelier "Plaine & Val de Sèvre", CNRS, F-79360 Villiers-en-Bois, France

AP, 0000-0003-2905-4769; FR, 0000-0003-1638-3141; YLC, 0000-0002-8466-5370; VB, 0000-0002-2320-7755; CA, 0000-0002-3045-2599

Assessing the various anthropogenic pressures imposed on honeybees requires characterizing the patterns and drivers of natural mortality. Using automated lifelong individual monitoring devices, we monitored worker bees in different geographical, seasonal and colony contexts creating a broad range of hive conditions. We measured their life-history traits and notably assessed whether lifespan is influenced by pre-foraging flight experience. Our results show that the age at the first flight and onset of foraging are critical factors that determine, to a large extent, lifespan. Most importantly, our results indicate that a large proportion (40%) of the bees die during pre-foraging stage, and for those surviving, the elapsed time and flight experience between the first flight and the onset of foraging is of paramount importance to maximize the number of days spent foraging. Once in the foraging stage, individuals experience a constant mortality risk of 9% and 36% per hour of foraging and per foraging day, respectively. In conclusion, the pre-foraging stage during which bees perform orientation flights is a critical driver of bee lifespan. We believe these data on the natural mortality risks in honeybee workers will help assess the impact of anthropogenic pressures on bees.

## 1. Introduction

Lifespan is a fundamental life-history trait that can exhibit tremendous variation between individuals of a given population. Variation in lifespan is generally linked to both intrinsic and

†Equal contribution.

extrinsic pressures acting separately or interacting with each other [1]. While intrinsic mortality risk is due to ageing (physical and functional degradation of the body), extrinsic mortality risk rather refers to environmental hazards (e.g. pollution, parasites, climate and predation).

Over the past decades, extrinsic pressures have greatly increased for many organisms due to anthropogenic alteration of ecosystems. This is especially true for insect pollinators [2–4], including the Western honeybee (*Apis mellifera*) [5,6], which is often used as a model species for pesticide risk assessment [7–9], but also for studying the consequences of decreased resource availability [10–13], climate change [14–16] and the spread of new biotic pressures [17–19]. In this context, it is critical to gather knowledge on the patterns and drivers of natural mortality in honeybees to help interpret the results of risk assessment studies.

Honeybees are social insects that exhibit striking caste-specific differences in longevity. While honeybee queens can live up to 5 years, workers usually only live two to six weeks in the summer and about 20 weeks in the winter [20]. The 10-fold difference between the summer and winter worker bee lifespan relies on differences in both intrinsic physiological senescence processes and extrinsic factors like exposure to environmental pressures (i.e. winter bees rarely leave the safe environment of the hive) [20]. In fact, as a result of age polyethism, the extrinsic mortality risk of summer bees is not constant throughout the individual's life [21,22]. Bees spend the first weeks of their adult life performing tasks within the hive environment, but then switch to foraging activity that exposes them to environmental hazards such as temperature, predation or dehydration. In addition, the transition to foraging activity is accompanied by a reduction in protein and lipid reserves [23], as well as the glycolipoprotein vitellogenin, a major antioxidant [24,25]. Foragers should then experience constant probability of death over time [26], but they may also face the depletion of limited glycogen reserves essential to their flight activity [27]. The age at onset of foraging (AOF), which can be modulated according to the size, demography and needs of the colony [28–30], as well as by several environmental factors [31–34], is, therefore, assumed to be an important driver of the lifespan of worker bees [35,36].

Foraging activity provisions the colony with floral resources which are essential for the survival and reproduction of the colony, and is typically preceded by a few days in which bees perform exploratory orientation flights [37] enabling them to learn the features of the hive entrance and the landscape around the hive. This pre-foraging stage allows bees to develop highly complex cognitive capacities essential to foraging, such as navigation and homing [38–40], as well as their flying capacity (sensory and motor performance) [41]. The experience developed during the pre-foraging stage might, therefore, determine their future performance as foragers and probably, their lifespan. As a matter of fact, one of the consequences of experimentally induced precocious foraging is the higher risk of death in the first foraging flights [35] and a shorter foraging span [36,42], possibly due to lower navigation capacities [43] (but see [36]), immature flight muscles [44,45] and/or a heavier body [46]. Last but not least, while performing learning flights, bees could be exposed to extrinsic mortality risks as suggested by Rueppell *et al.* [22] and Requier *et al.* [47]. Therefore, there may be a trade-off between the risks associated with the performance of pre-foraging flights and the benefits from accumulating flight experience during this stage, which might translate into higher performances and survival at the foraging stage.

Our aim was to document this particular and critical period of learning flight. For that purpose, we monitored, from emergence to death, the in-and-out activity of 3786 worker honeybees using automated lifelong individual monitoring devices. This was done in a broad range of conditions, consisting of multiple geographical, seasonal and colony contexts, to evaluate diverse life histories related to the well-known behavioural plasticity in honeybees [48]. Variability in environmental contexts was notably achieved by tracking bees originating from different source colonies each month from April to September in two different geographical sites in France. We then measured their life-history traits and, particularly, identified AOF based on the detection of ontogenetic changes in time-activity budget series of individual bees [47]. The overarching aim was to assess whether lifespan is influenced by flight experience, in order to better characterize the patterns and drivers of natural mortality risk in free-range honeybees. For this purpose, we evaluated (i) to what extent the pre-foraging stage is associated with mortality risks, and (ii) how this pre-foraging stage might influence the duration of the foraging stage.

# 2. Material and methods

## 2.1. Geographical, seasonal and colony contexts of monitored beehives

To date, two main techniques are available to monitor the in-and-out activity of honeybees at the beehive entrance, the radio-frequency identification device (RFID) [49] and the optical counter based on image

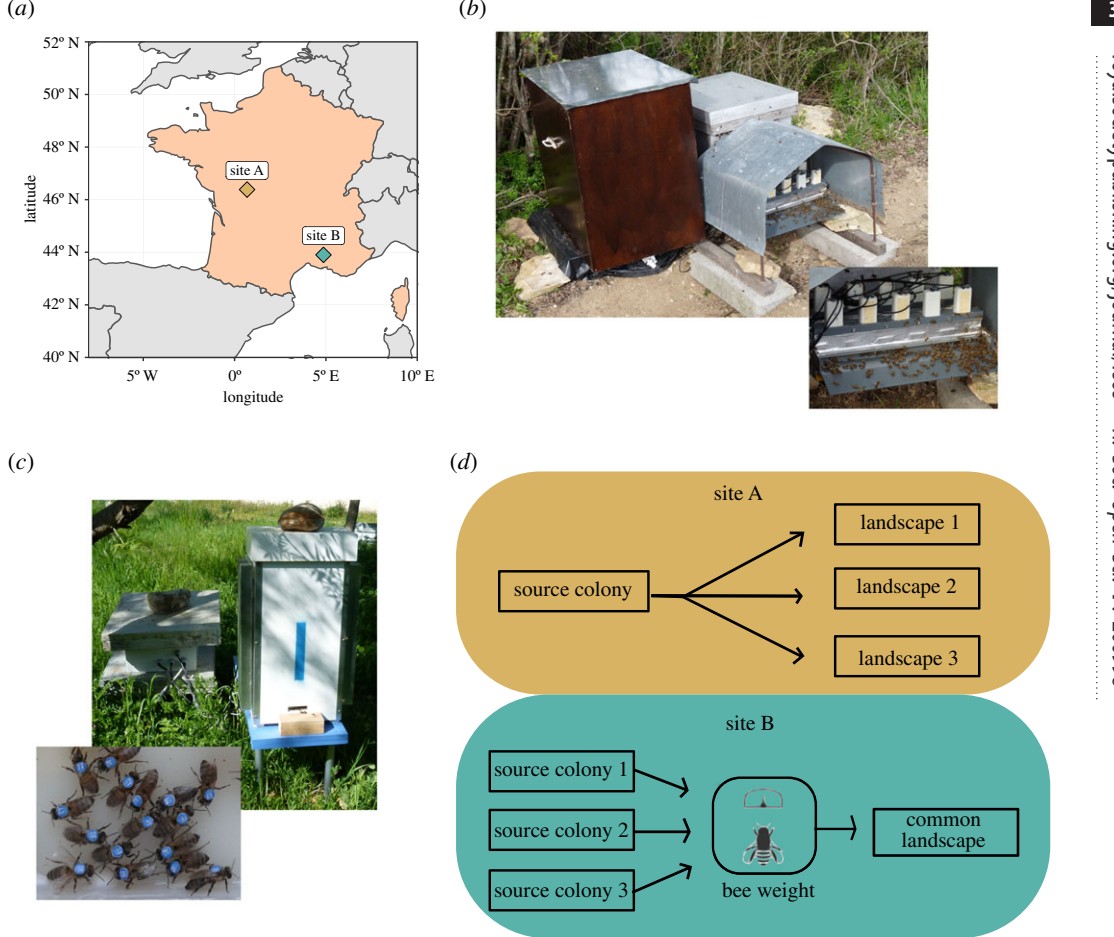

**Figure 1.** Experimental design. (*a*) Map of France indicating the geographical position of site A (Chizé) and site B (Avignon). (*b*) RFID system used at site A. (*c*) Optical counter system used at site B. (*d*) Experimental design.

identification of barcode tags [50,51]. With such devices, the time-activity budgets of flights (duration and number of trips per day) are similarly recorded at the individual level. We equipped three colonies with RFID in 2011 in the Long Term Social-Ecological Research 'Zone Atelier Plaine & Val de Sèvre' (LTSER ZA-PVS) [52] in central western France (46°23′ N, 0°41′ W, site A, figure 1), and one colony with image-based optical counters in 2018 at the National Institute for Agriculture, Food and Environment of Avignon in southeastern France (43°54′ N, 4°52′ E, site B, figure 1). The RFID device consists of two adjacent rows of five contiguous RFID readers (iID2000, 2k6 HEAD; Microsensys GmbH, Erfurt, Germany) placed at the entrance of the hives (figure 1). The image-based optical counter comprises a camera that monitors the hive entrance and image analysis software that detects and registers the barcode [50,51]. These two geographical areas vary in climate context (Oceanic for site A and Mediterranean for site B), giving different colony dynamics (e.g. different timing of peak brood production).

Beyond this inter-regional variability, we further introduced intra-regional variations in bee life-history traits (figure 1). To do so, cohorts of bees were tagged each month from April to September in both sites to add seasonal variation. In addition, we simulated a third factor of variation by manipulating the environment of tagged bees (figure 1). In site A, all tagged bees came from a single source colony (*A. mellifera mellifera × caucasica* strain) but were introduced into three monitored beehives (common 10-frame Dadant–Blatt model with sister queen of *A. mellifera mellifera × caucasica* strain) placed about 30 km from the source location. Moreover, the three monitored colonies were separated from each other by about 15 km to provide independent foraging landscapes of the tagged bees. In site B, tagged bees that were introduced each month in a single monitored colony came from three source colonies (*A. mellifera mellifera × ligustica* strain for the source and monitored colonies). Hence bees could be identified based on their colony of origin. These two different scenarios allowed the creation of variability in environmental and colony contexts with potential effects on bee life history.

## 2.2. Lifelong individual monitoring of bees

To monitor individuals throughout their entire adult life, we collected newly emerged bees. Bees were selected from brood frames containing late-stage pupae, all adult bees were removed from this frame and placed in an incubator at 34°C and 50–70% humidity. Newly emerged bees were then collected after 2–10 h of incubation and tagged. In site A, tags consisted of RFID microchips of $1.0 \times 1.6 \times 0.5$ mm (mic3—TAG 64 bit RO, iID2000, 13.56 MHz system, Microsensys GmbH, Erfurt, Germany). RFID tags weighed approximately 3 mg, i.e. 3% of an adult honeybee's body mass. This weight is considered low enough to not interfere with the individual life and tasks [49]. In site B, tags consisted of data-matrix barcodes (3 mm diameter) printed on laminated paper. Both tag types were glued on the bee thorax using biocompatible dental cement (TempoSIL2) in site A and Sader glue in site B. Because bee weight might influence flying capabilities [46], in site B, all newly emerged bees were weighed individually before tagging in order to assess the potential influence of body weight at emergence on bee lifespan. Indeed, bee weight at emergence might vary according to their nutritional state during larval development [53].

Tagged bees (less than 24 h old) were then introduced into colonies equipped with monitoring systems and monitored until death in both sites. All colonies were adequately treated against the mite *Varroa destructor* each fall and no visible disease symptoms were observed.

## 2.3. Measuring bee life-history traits

Of the 2100 and 1686 individually marked bees in sites A and B, respectively, life-history data were successfully recorded for 1867 (89%) and 1507 (89%) bees, i.e. with at least one exit and entrance sequence. This 11% loss (identical in the two sites) is attributable to the following: loss of barcode/ RFID tag prior to leaving the hive, rejection from the host colony by nest-mates or death during the bee's first flight. Bees without at least one exit and entrance sequence were excluded from analyses, thus 3374 bees were available for the analyses. For the analysis of flight activity, exits with durations shorter than 2 s or longer than 180 min were excluded (not considered as flights). We then calculated the age of first exit and entrance sequence (AFE, in days), the lifespan (i.e. the age at last exit) and AOF for each bee using the *aof* function developed in the *aof* R-package [47,54]. The *aof* function is a simple statistical procedure, derived from the behavioural change point analysis approach [55], a well-appreciated technique of likelihood comparisons to statistically determine behavioural changes. It aims to detect a single behavioural change in univariate time series. The *aof* function works at the individual level thus accounting for inter-individual variation to detect, assess and quantify shifts in the temporal pattern of time-activity budgets recorded by individual lifelong monitoring. Indeed, a clear breakpoint in duration, frequency and time of occurrence in the day was found between learning and foraging flights of honeybees, so-called AOF [47]. Thus, AOF is estimated from three daily activities, (i) the number of exit and entrance sequences per day, (ii) the duration of these sequences, and (iii) the hour of the day that they were performed. Then, these three values are averaged to give an AOF estimate in days. The *aof* function is freely available through the *aof* R-package that includes documentation, source code and examples using both simulated and empirical dataset [47,54].

## 2.4. Data analysis

### 2.4.1. Pre-foraging flight parameter effects on lifespan

To assess which factors were associated with lifespan we used Cox proportional-hazard models for datasets of site A and site B separately (given the differences in study sites and monitoring devices). First, we tested for autocorrelation of the residuals using a Durbin–Watson test on a linear regression of lifespan as a function of AFE, AOF and cohort. Once independence of the residuals had been established, we used the Cox model to investigate the association between bee survival and the predictors. Both models included AFE and AOF as predictors. The site A model included the landscape as a predictor and the replicate (monthly cohort) as a grouping cluster. The site B model included bee weight as a predictor and the replicate as a grouping cluster.

To assess the distribution of lifespan frequency, we used Hartigan & Hartigan's dip test for unimodality [56]. The dip test measures multimodality in a sample by estimating the maximum difference, over all sample points, between the empirical distribution function and the unimodal distribution function. All data analyses were performed in R v. 3.5.2 [57].

**Table 1.** Life-history traits of worker honeybees. Nt, number of tagged bees; N, number of analysed bees (at least one exit and entrance sequence), AFE, age at first exit; AOF, age at onset of foraging; LSP, lifespan; s.d., standard deviation (days). Non-forager and forager bees correspond to bees that did not reach the AOF and those that did, respectively.

| | Nt | N | all bees | | | non foragers | foragers |
| | | | AFE | AOF | LSP | LSP | LSP |
|---|---|---|---|---|---|---|---|
| site A | | | | | | | |
| April | 300 | 250 | 11.4 ± 4.3 | 19.2 ± 5.1 | 18 ± 9.9 | 11.5 ± 5.4 | 26.9 ± 7.6 |
| May | 300 | 274 | 10 ± 5.3 | 16.4 ± 8.1 | 15.4 ± 9.9 | 10.6 ± 5.8 | 23.5 ± 10.2 |
| June | 150 | 129 | 10.8 ± 5 | 19.7 ± 6.1 | 19.1 ± 11.6 | 11 ± 6.8 | 28.7 ± 8.2 |
| July | 450 | 420 | 17.8 ± 8.5 | 31.4 ± 12.4 | 30.5 ± 17.1 | 17.2 ± 10 | 41 ± 14 |
| August | 450 | 407 | 11 ± 5.7 | 20.3 ± 7.2 | 20.2 ± 12.3 | 11.3 ± 7.6 | 28.9 ± 9.6 |
| September | 450 | 387 | 9.7 ± 5.3 | 18.8 ± 8.6 | 18.7 ± 12.1 | 12.5 ± 8.9 | 27.8 ± 10.1 |
| site B | | | | | | | |
| April | 300 | 260 | 7.2 ± 3.3 | 19.2 ± 4.1 | 20.1 ± 10.8 | 9.3 ± 5.8 | 27.8 ± 5.8 |
| May | 300 | 276 | 8.5 ± 4.5 | 19 ± 4.6 | 20.5 ± 9.3 | 11 ± 6.2 | 25.9 ± 5.7 |
| June | 300 | 247 | 9.3 ± 5.6 | 24.5 ± 8.5 | 16.8 ± 14.1 | 8.3 ± 5 | 33.2 ± 11.2 |
| July | 300 | 272 | 7.8 ± 4.2 | 19.8 ± 5.6 | 20.1 ± 10 | 11.7 ± 7.5 | 26.6 ± 6 |
| August | 300 | 269 | 10.1 ± 4.6 | 19.8 ± 5 | 18 ± 9.8 | 11.6 ± 6.1 | 26.7 ± 6.7 |
| September | 200 | 183 | 8.5 ± 3.9 | 15.2 ± 4.3 | 16.7 ± 7.2 | 13.3 ± 6.2 | 21.7 ± 5.5 |

### 2.4.2. Pre-foraging effects on foraging stage

We monitored the survivorship of forager honeybees and tested whether forager survivorship followed a type II log-linear curve with a constant mortality rate, as previously observed by Visscher & Dukas [26]. For this purpose, we calculated the Nelson–Aalen cumulative hazard estimate for four life-traits: foraging stage (number of days from the AOF to death), foraging tenure (actual days of foraging), cumulative minutes foraged and the cumulative number of flights. To assess the overall hazard of foraging, we used the slopes of the linear regression of the cumulative hazard estimates [58]. Linear regressions were performed with a minimum sample size of 24 bees (other samples with fewer bees were excluded) and the normal distribution of residuals was controlled using the Shapiro–Wilk test.

We further analysed whether pre-foraging traits influence the foraging stage. We used a generalized linear mixed-effect model (GLMM) on the square root of the foraging stage for site A and site B separately (see above). The GLMMs had a Gaussian distribution of errors that considered AFE, AOF, the foraging intensity (mean min spent per day foraging), source colony (site B dataset), weight at emergence (site B dataset) and landscape (site A dataset) as the fixed effects. The cohort was considered as a random effect ($a_i$) and the variance was allowed to differ among the six different cohorts ($\epsilon_j$).

$$\sqrt{\text{Foraging stage}} = \propto + \beta_1(\text{AFE}) + \beta_2(\text{AOF}) + \beta_3(\text{intensity}) + \beta_4(\text{source colony}) + \beta_5(\text{weight}) + a_i + \epsilon_j.$$

## 3. Results

### 3.1. Pre-foraging stage effects on lifespan

From April to the end of September, the average lifespan of bees was $21.2 \pm 13.9$ days (standard deviation, $n = 1867$ bees) and $18.9 \pm 10.6$ days ($n = 1507$ bees) in site A and site B, respectively. Lifespan was, therefore, highly variable (table 1). At both sites, a type III survivorship curve was observed with a high early mortality (figure 2).

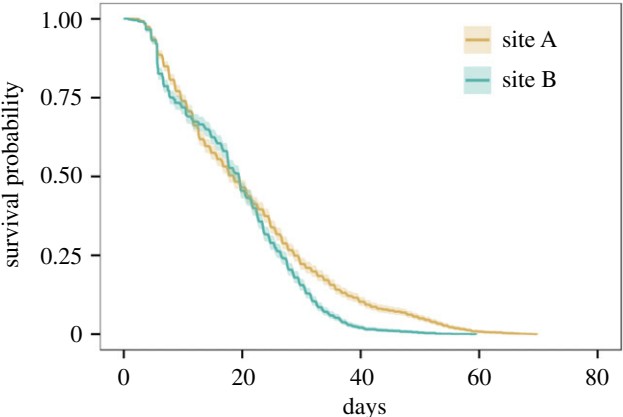

**Figure 2.** Modelled survival probabilities of worker bees in site A and site B with Cox proportional-hazard models.

**Table 2.** Cox proportional-hazard model results on the influence of AFE, AOF, landscape, source colony and bee weight on bee survival.

|  | coefficient | s.e. | Z | p |
|---|---|---|---|---|
| **site A** |  |  |  |  |
| AFE | 0.044 | 0.007 | 5.54 | <0.001 |
| AOF | −0.253 | 0.009 | −16.75 | <0.001 |
| landscape 2 | 0.322 | 0.102 | 1.5 | 0.135 |
| landscape 3 | 0.348 | 0.101 | 1.968 | 0.049 |
| **site B** |  |  |  |  |
| AFE | 0.056 | 0.009 | 9.601 | <0.001 |
| AOF | −0.237 | 0.009 | −5.484 | <0.001 |
| source colony 2 | −0.16 | 0.089 | −1.286 | 0.198 |
| source colony 3 | −0.078 | 0.096 | −0.683 | 0.495 |
| weight | 0.002 | 0.005 | 0.541 | 0.588 |

The Cox proportional-hazard model for site A shows that landscape did not significantly affect honeybee survival (site A, table 2), nor did the weight of the individual at emergence and the source colony at site B (table 2). At both sites, the major drivers of honeybee lifespan were age at the first exit (AFE) and the age at the onset of foraging (AOF) (table 2). The average AFE was $12.2 \pm 6.9$ days at site A and $8.5 \pm 4.5$ at site B. The average AOF was $22.4 \pm 10.6$ days at site A and $19.6 \pm 5.7$ at site B. The replicate of July at site A had the highest average AOF of $30.5 \pm 17.1$ days (table 1). Surprisingly, the average AOF was slightly higher than the average bee lifespan in both sites, reflecting a high number of bees that die early and never become foragers (see below). AOF and AFE were positively correlated (Spearman's $\rho = 0.68$ and 0.26 at sites A and B, respectively) but had opposite effects on mortality risk in both models (table 2). AOF had a strong negative effect: as AOF increased, mortality risk decreased. AFE had a softer and opposite effect showing that as AFE increased, mortality risk also increased.

In both sites, we observed a bimodal distribution in the lifespan of honeybee workers (Hartigan & Hartigan's dip test for unimodality, $p < 0.05$ for all cohorts, except June and July at site A; figure 3). A first peak in mortality was observed around the age of 10 days, while the second peak occurred after the age of 20 days. According to the AOF function, these two populations of bees (one dying earlier than the other) could be segregated based on their behaviour; the first population corresponding to those who did not reach the AOF and the second one corresponding to the true foragers (reaching the AOF). The proportion of bees that never reached the AOF was 53.6% and 49.8% at sites A and B, respectively, which could partly be explained by the proportion of bees that died before the age of 15 days (early mortality), which was 42.4% and 37.5% at sites A and B, respectively (figure 4).

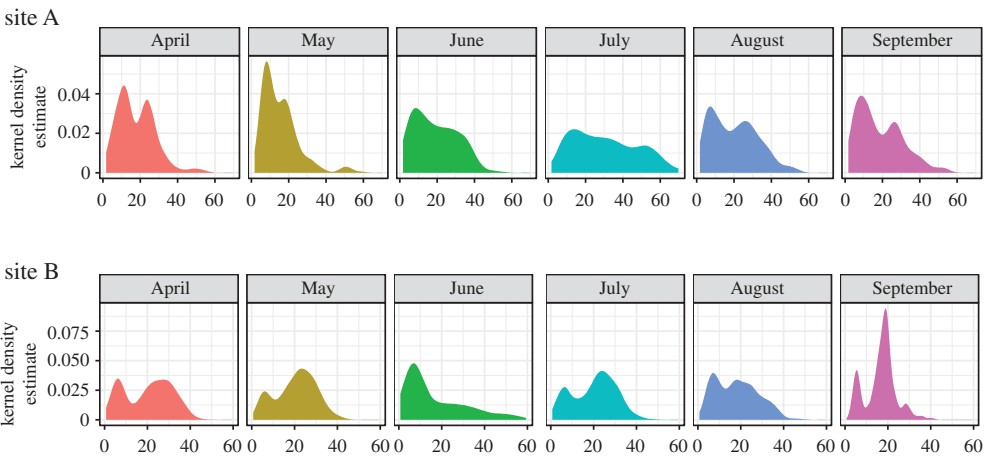

**Figure 3.** Kernel density estimation of worker bee lifespan. Distribution of bee lifespan is shown for each month from April to September in sites A and B. Hartigan & Hartigan's dip test for unimodality: $p < 0.05$ for each month except for June and July in site A ($p = 0.71$ and $p = 0.39$, respectively).

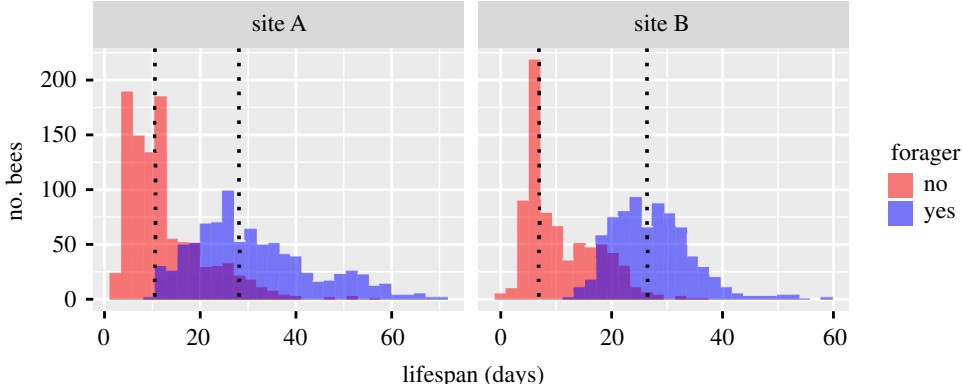

**Figure 4.** Histograms of the lifespan of non-forager (red) and forager bees (blue) at both experimental sites (sites A and B). Dashed lines represent the median lifespan of non-forager and forager bees.

## 3.2. Pre-foraging effects on foraging stage

The survivorship of foragers followed a type II log-linear curve. The regressions of the cumulative hazard on the foraging stage (number of days between AOF and death) (Shapiro–Wilk test $W = 0.82$ and $p < 0.0001$, $W = 0.82$ and $p < 0.0001$, for sites A and B, respectively) and on the cumulative number of flights ($W = 0.95$ and $p < 0.001$, $W = 0.86$ and $p < 0.0001$, for sites A and B, respectively) did not present a normal distribution of their residuals so they were not considered. However, the cumulative hazard of foraging (Neslon–Aalen estimates) could be explained as a function of the foraging tenure in both sites ($W = 0.88$ and $p = 0.07$, $W = 0.86$ and $p = 0.052$, for sites A and B, respectively; figure 5) and cumulative time spent foraging ($W = 0.99$ and $p = 0.12$) at site A. The residual distribution of cumulative time spent foraging was close to normality at site B but failed to pass the test ($W = 0.98$ and $p = 0.02$; figure 5). The hazard associated with foraging, as interpreted from the slopes of these regressions indicate bees were exposed to a constant mortality rate of 32% and 40% per foraging day (foraging tenure) at sites A and B, respectively (figure 5a) and 8.7% per every hour of foraging at site A (figure 5b; for information, in site B the mortality rate was 9.9%).

The foraging stage of honeybees ranged from 1 to 31 days. The average length of the foraging stage at sites A and B was 8.7 ± 4.1 and 7.4 ± 3.3 days, respectively. The landscape did not influence the foraging stage at site A; neither did the source colony and weight at emergence at site B (table 3). In site B, variations in the foraging stage were slightly but significantly explained by the foraging intensity, defined as the number of minutes foraged divided by the number of foraging days; bees who exhibited a high foraging intensity tended to have a longer foraging stage (table 3).

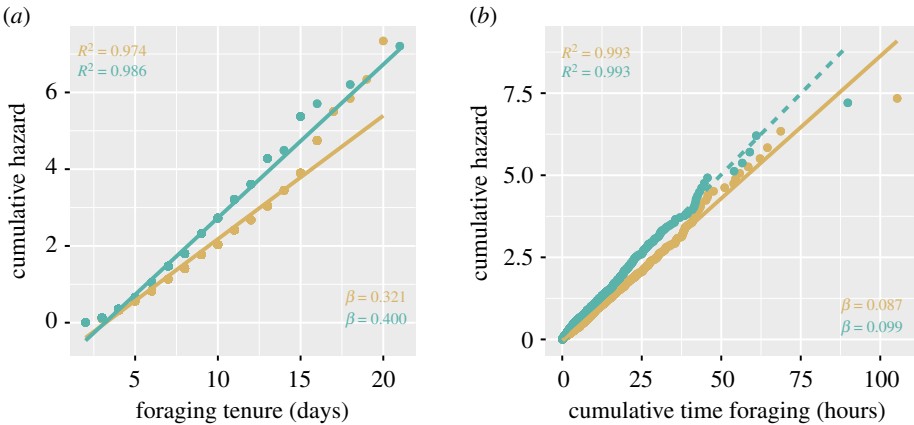

**Figure 5.** Nelson–Aalen estimate of the cumulative hazard of foraging as measured by (*a*) the foraging tenure and (*b*) the cumulative time spent foraging in site A (gold) and site B (green). Dotted line shows model with abnormal distribution of residuals.

**Table 3.** Mixed effect model results on the influence of AFE, AOF, foraging intensity, source colony, bee weight and landscape on duration of the foraging stage.

|  | coefficient | s.e. | d.f. | *t*-value | *p*-value |
|---|---|---|---|---|---|
| site A |  |  |  |  |  |
| intercept | 2.4064 | 0.0823 | 854 | 29.23 | <0.001 |
| AFE | −0.0252 | 0.0042 | 854 | −5.94 | <0.001 |
| AOF | 0.0350 | 0.0030 | 854 | 11.63 | <0.001 |
| intensity | 0.0004 | 0.0003 | 854 | 1.19 | 0.2351 |
| landscape 2 | −0.0125 | 0.0636 | 854 | −0.19 | 0.8438 |
| landscape 3 | −0.0723 | 0.0643 | 854 | −1.13 | 0.2570 |
| site B |  |  |  |  |  |
| intercept | 2.5379 | 0.3384 | 735 | 7.49 | <0.001 |
| AFE | −0.0308 | 0.0044 | 735 | −6.94 | <0.001 |
| AOF | 0.0261 | 0.0038 | 735 | 6.82 | <0.001 |
| intensity | 0.0014 | 0.0002 | 735 | 4.96 | <0.001 |
| source colony 2 | 0.0447 | 0.0429 | 735 | 1.04 | 0.2981 |
| source colony 3 | −0.1226 | 0.0469 | 735 | −2.61 | 0.0092 |
| weight | −0.0018 | 0.0029 | 735 | −0.65 | 0.5147 |

However, at both sites, small AFE and high AOF maximized the foraging stage (table 3). This suggested that the number of days between AFE and AOF played an important role in determining the foraging stage. This was verified by the positive correlation between the duration of this learning (pre-foraging) stage and the foraging stage ($r^2 = 0.31$, $r^2 = 0.38$ for sites A and B, respectively). The number of pre-foraging flights and the amount of minutes accumulated during the pre-foraging flights were slightly associated with a higher foraging stage at both sites (table 4).

Finally, among the different parameters of the pre-foraging stage, the minutes accumulated during the pre-foraging flights had the highest correlation coefficient with the foraging intensity; three times higher than for the time elapsed between AFE and AOF (table 4).

## 4. Discussion

By analysing two large datasets of honeybee flight activity, obtained in two different geographical areas and years, we found a consistent influence of the pre-foraging stage (as measured by the time between

**Table 4.** Pearson's correlation coefficients between pre-foraging experience and foraging stage duration and intensity.

| | duration of the foraging stage | | foraging intensity | |
|---|---|---|---|---|
| | coefficient | *p*-value | coefficient | *p*-value |
| site A | | | | |
| number of pre-foraging flights | 0.22 | <0.001 | 0.27 | <0.001 |
| cumulative minutes of pre-foraging flights | 0.21 | <0.001 | 0.46 | <0.001 |
| number of days between AFE and AOF | 0.38 | <0.001 | 0.12 | <0.001 |
| site B | | | | |
| number of pre-foraging flights | 0.17 | <0.001 | 0.35 | <0.001 |
| cumulative minutes of pre-foraging flights | 0.16 | <0.001 | 0.48 | <0.001 |
| number of days between AFE and AOF | 0.31 | <0.001 | 0.14 | <0.001 |

the age at the first exit and the age at the onset of foraging) on bee lifespan and foraging performance. Specifically, the pre-foraging stage is associated with high mortality risks but contributes to shaping the individual's foraging performance and duration.

During this pre-foraging stage, we observed a large portion of the worker bees dying; 40% of bees died before the age of 15 days and 50% of bees never reached the foraging stage. In fact, the distribution of bee lifespan was mostly bimodal, with a short-lived population of worker bees that never became foragers, and a second population of bees living longer and becoming foragers. This high mortality of bees occurring before they become foragers was previously noticed [22] (although not characterized). It indicates that orientation flights and the transition to foraging represent a high risk to bees. The individuals that survive this pre-foraging stage are the bees that then become true foragers. It is not clear which factors determine a successful transition to foraging tasks and why only half of the population survive and becomes foragers. It is possible that some bees became foragers without exhibiting a clear transition to foraging activity (AOF) and hence were not detected by the *aof* procedure [47,54]. However, this limitation in our methodology does not explain the high early mortality that can only be explained by the extrinsic mortality risks associated with orientation and cleansing flights (e.g. predation, weather) and/or poor learning abilities during the first flights performed during their life. The strong loss in the worker bee population could help explain the previously described elite worker bees that perform a disproportionate amount of the foraging for the colony [48,59]. Indeed, both studies found that not all bees contribute equally to the total colony activity, but most of the flight activity is done by a small number of individuals: 12 and 20% of bees performed 50% of the colony flight activity [48,59]. Even though this skew in activity is flexible and may depend on colony needs, the origin of this inter-individual variability among honeybee foragers is not well understood. It could be attributed to genetically based differences in neural and metabolic functions, to bee health status before foraging activities, or to the increase in foraging performance (e.g. number of daily flights) as a bee gains experience [59]. Our data suggest that the existence of a large population of short-lived bees (with, therefore, a minor contribution to foraging activity) could also explain this skew in colony activity.

As expected, of the behavioural parameters we have evaluated, AOF clearly influences the bee lifespan. Indeed, when bees switch to foraging tasks, they get chronically exposed to extrinsic mortality risks. Therefore, the later the AOF is, the longer bees are expected to live. In addition, bees that start foraging too early generally exhibit a reduced foraging life [36], probably because they are not optimally prepared for the foraging tasks as compared to normal-aged foragers [35]. This was further confirmed by the positive influence of AOF on bee foraging stage. Conversely, the earlier the AFE was, the lower the risk of death and longer the foraging stage. The correlation between AFE and foraging stage was previously observed by Rueppel *et al.* [28] (in their study what they reported as AOF is probably AFE, since they made no distinction between them). This seems paradoxical given that the onset of flight activity should normally be associated with increased extrinsic mortality risks, and it is not clear why early AFE would improve foraging stage. However, it suggests that the longer the time lapse between AFE and AOF, the longer the bee will forage. This is interesting because

during this period bees mostly perform orientation flights to learn and develop a spatial memory of the landscape around the hive [38–40]. Such a pattern was also observed with honeybee drones: the time that they spent on orientation flights was positively correlated to the time that they spent on mating flights [60].

Several studies have shown that foragers spend a considerable portion of their lifespan learning and improving their foraging skills through experience [44,59,61]. They gradually increase their foraging performance over a period of more than a week, and then, performances reach a plateau [59] or decrease [61]. Our data suggest that the pre-foraging stage is also of paramount importance to maximize the foraging stage. It, therefore, seems that the more days that they spend accumulating experience, the better their future foraging performance. As a matter of fact, experimentally induced precocious foragers exhibit some deficit in developing spatial memory as compared to normal-aged foragers [43]. It is also possible that a consequent pre-foraging activity is required not only to optimize cognitive functions but also physiological functions associated with flight capacities, like a decrease in body mass [46], an increase in cytochrome concentrations [62], thoracic glycogen levels [63], citrate synthase levels and troponin T 10A expression [44], which yield a strong increase in flight metabolic rate [64,65]. The physiological maturation hypothesis seems supported by the fact that the foraging stage is more strongly correlated to the number of days between AFE and AOF than by the actual amount of pre-foraging experience (number and minutes of pre-foraging flights). In turn, the amount of pre-foraging experience was more strongly correlated to the foraging intensity, suggesting again a positive influence of pre-foraging activity on future foraging skills probably via learning and/or physiological maturation. Nevertheless, manipulative experiments would be needed to understand how precisely the pre-foraging stage influences foraging.

Honeybees have contributed to show that ageing may be a flexible process and does not necessarily result from an inevitable decline of physiological functions [66,67]. Our results on the connection between pre-foraging experience and forager lifespan further supports this view and the decoupling between mortality and chronological age [68]. This represents another level of plasticity in the regulation of lifespan, besides the well-known stage-dependent trajectory of ageing (nurse versus forager) [66].

Surprisingly, the foraging intensity (mean number of min spent per day foraging) had a small but positive effect on the foraging stage in site B. This goes against the rate-of-living theory, according to which, the higher an organism's metabolism, the shorter its lifespan [69], as well as the hypothesis of a limited budget of energy expenditure determining forager lifespan [27]. However, this could be explained by the experience gained by bees during their foraging trips, which contribute to improving their foraging performance [44,59,61]. Foragers, by improving their foraging performance (learning and navigation abilities), might then be able to sustain longer foraging activity.

Finally, we found that the main mortality factor of foragers was not associated with senescence (type I survivorship curve), or high early mortality (type III survivorship curve), but rather forager bees faced a constant probability of death during their foraging activity (type II curve). This confirms previous work based on the visual recording of 33 individuals [26], as well as the decoupling between mortality and functional senescence in forager bees [68]. Our data generated with automated monitoring systems allowed us to further understand this constant probability of mortality, as it was best explained by the cumulative time spent foraging and foraging tenure than by the foraging stage and the cumulative number of foraging trips. Foragers were exposed to an average 9% and 36% probability of death per hour and day of foraging, respectively, which indicates that this is the risk associated with actual foraging flights (the time spent in the field) that is critical to forager survivorship. The risk mortality during foraging is generally assumed to be due to predation and weather conditions but it would be interesting to determine how robust these death rates are and whether they can change according to predation, weather or other factors, as they were quite consistent across our experimental set-ups. Furthermore, our results on honeybee lifespan (23–41 days) and foraging span (approximately 8 days) are in agreement with those reported by Fukuda and Sekiguchi for spring and summer bees more than 50 years ago (lifespan between 19 and 47 days) [70] and by Dukas & Visscher [71] (foraging span of around 7.7 days) 25 years ago. This is interesting, since honeybees probably did not face the same anthropogenic pressures (e.g. pesticide exposure) as they do now.

# 5. Conclusion

In conclusion, the pre-foraging stage is highly critical to honeybee survival. During this transition, bees learn their environment (orientation flights) and develop a physiology adapted to flight, but also

get exposed to extrinsic mortality risks (predation and weather conditions). This leads to a cost for the colony with a sharp reduction in the forager population. However, this learning stage is essential to accumulate flight experience and improve foraging performance and duration. Another aspect of our study is the remarkable consistency of results between the two datasets obtained in different environmental conditions and years, and using two different automatic monitoring systems, highlighting the robustness of the pre-foraging influence on bee lifespan. Finally, we hope the improved characterization of bee mortality pattern (e.g. bimodal distribution of bee lifespan, % of bees reaching the AOF, and forager death probability), will inform colony dynamic simulations [72–74] and future experimental studies to better determine the effects of the growing anthropogenic pressures on honeybees.

Ethics. All honeybees were handled with care in this study.

Data accessibility. The datasets supporting this article are available at the Dryad Digital Repository: https://doi.org/10.5061/dryad.s7h44j154 [75].

Authors' contributions. A.P. and F.R. contributed equally to this work. A.P., F.R., C.A. and V.B. conceived and designed the study. A.P., F.R. and D.C. collected the data. A.P. and F.R. performed the data analyses. A.P, F.R. and C.A. wrote the paper with suggestions from Y.L.C. and V.B.

Competing interests. The authors of this manuscript declare no competing interests.

Funding. This work was supported by grants from the AgreenSkills Fellowship Program, the French Ministry of Agriculture (CASDAR, POLINOV project no. 9535), the Poitou-Charentes Region, the European Community programme (797/2004) for French beekeeping coordinated by the French Ministry of Agriculture (TECHBEE project).

Acknowledgements. Special thanks to Pierrick Aupinel, Axel Decourtye and Jean-François Odoux for logistical help carrying out fieldwork in central western France. We also thank Clovis Toullet and Charlotte Ruger for field assistance, and Cynthia McDonnell for the English editing.

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
