## [Reviewer comments · Royal Society Open Science]

Review History

RSOS-200998.R0 (Original submission)

Review form: Reviewer 1

Is the manuscript scientifically sound in its present form?

Yes

Are the interpretations and conclusions justified by the results?

Yes

Is the language acceptable?

Yes

Do you have any ethical concerns with this paper?

No

Have you any concerns about statistical analyses in this paper?

No

Recommendation?

Accept with minor revision (please list in comments)

Comments to the Author(s)

This ms provides very important descriptive data on honey bee behavior, delivered with new automated monitoring technology. The data will be very useful to a variety of researchers from both applied and basic perspectives.

The study is robust with excellent sample size of bees and colonies, and also exposure to different environmental conditions. The consistency of the results across these levels of analysis speaks to robust results.

I have the following suggestions to improve the manuscript.

It is not clear how age at onset of foraging is calculated. The information apparently is in an R package, cited herein. It is recommended that the authors add a sentence on what assumptions were made in terms of activity changes to trigger classification as foraging flight, and ultimately as forager. For example, how long was a round trip classified as a foraging vs. orientation trip? This is an issue that investigators pay close attention to. There is general consensus, so it would be good to know whether this study uses similar metrics.

The ms does a good job of citing a lot of the relevant literature, but has ignored some of the earlier literature that used manual observation to arrive at some of the same conclusions, concerning the relationship between age at onset of flight and lifespan. This would include papers by Kolmes, Winston and Robinson from the 1980's. Indeed, some of the similarities in duration of foraging period would be very interesting to point out as bees 40 years ago did not face the same exposure to parasitic mites, pesticides, poor nutrition, and pathogens.

My second suggestion is that the discussion could consider the implication of these results in light of key theories of aging and the evolution of senescence. Honey bees have figured prominently in the development of theories in this area, and I sense that the results here would make a nice contribution to some of those considerations. I understand this needs to be a short paper, so perhaps the authors can pick just a single salient aspect to comment on.

Decision letter (RSOS-200998.R0)

Dear Dr Prado

The Editors assigned to your paper RSOS-200998 "Honey bee lifespan: the critical role of pre-foraging stage" have now received comments from reviewers and would like you to revise the paper in accordance with the reviewer comments and any comments from the Editors. Please note this decision does not guarantee eventual acceptance.

Please submit your revised manuscript and required files (see below) no later than 21 days from today's (ie 03-Sep-2020) date. Note: the ScholarOne system will 'lock' if submission of the revision is attempted 21 or more days after the deadline. If you do not think you will be able to meet this deadline please contact the editorial office immediately.

on behalf of Prof Kevin Padian (Subject Editor)
openscience@royalsociety.org

Associate Editor Comments to Author:

Associate Editor: 1

Comments to the Author:

Please fully respond to the queries/concerns of the reviewer

Editor comments:

Thanks for your submission. We have only one reviewer who likes the manuscript but has some concerns that would suggest a bit of re-writing and perhaps reconsideration. Please attend to these and we look forward to your next version.

Reviewer comments to Author:

Reviewer: 1

Comments to the Author(s)

This ms provides very important descriptive data on honey bee behavior, delivered with new automated monitoring technology. The data will be very useful to a variety of researchers from both applied and basic perspectives.

The study is robust with excellent sample size of bees and colonies, and also exposure to different environmental conditions. The consistency of the results across these levels of analysis speaks to robust results.

I have the following suggestions to improve the manuscript.

It is not clear how age at onset of foraging is calculated. The information apparently is in an R package, cited herein. It is recommended that the authors add a sentence on what assumptions were made in terms of activity changes to trigger classification as foraging flight, and ultimately as forager. For example, how long was a round trip classified as a foraging vs. orientation trip? This is an issue that investigators pay close attention to. There is general consensus, so it would be good to know whether this study uses similar metrics.

The ms does a good job of citing a lot of the relevant literature, but has ignored some of the earlier literature that used manual observation to arrive at some of the same conclusions, concerning the relationship between age at onset of flight and lifespan. This would include papers by Kolmes, Winston and Robinson from the 1980's. Indeed, some of the similarities in duration of foraging period would be very interesting to point out as bees 40 years ago did not face the same exposure to parasitic mites, pesticides, poor nutrition, and pathogens.

My second suggestion is that the discussion could consider the implication of these results in light of key theories of aging and the evolution of senescence. Honey bees have figured prominently in the development of theories in this area, and I sense that the results here would make a nice contribution to some of those considerations. I understand this needs to be a short paper, so perhaps the authors can pick just a single salient aspect to comment on.

===PREPARING YOUR MANUSCRIPT===

- one version identifying all the changes that have been made (for instance, in coloured highlight, in bold text, or tracked changes);
- a 'clean' version of the new manuscript that incorporates the changes made, but does not highlight them. This version will be used for typesetting if your manuscript is accepted.

===PREPARING YOUR REVISION IN SCHOLARONE===

Please ensure that you include a summary of your paper at Step 2 'Type, Title, & Abstract'. This should be no more than 100 words to explain to a non-scientific audience the key findings of your

research. This will be included in a weekly highlights email circulated by the Royal Society press office to national UK, international, and scientific news outlets to promote your work.

Author's Response to Decision Letter for (RSOS-200998.R0)

See Appendix A.

RSOS-200998.R1 (Revision)

Review form: Reviewer 1

Is the manuscript scientifically sound in its present form?

Yes

Are the interpretations and conclusions justified by the results?

Yes

Is the language acceptable?

Yes

Do you have any ethical concerns with this paper?

No

Have you any concerns about statistical analyses in this paper?

No

Recommendation?

Accept as is

Comments to the Author(s)

This ms has been revised nicely to address earlier concerns.

Decision letter (RSOS-200998.R1)

Dear Dr Prado,

It is a pleasure to accept your manuscript entitled "Honey bee lifespan: the critical role of pre-foraging stage" in its current form for publication in Royal Society Open Science. The comments of the reviewer(s) who reviewed your manuscript are included at the foot of this letter.

===COVID-SPECIFIC TEXT -- WILL ONLY BE ADDED TO COVID-PAPERS BY THE EDITORIAL OFFICE===

COVID-19 rapid publication process:

We are taking steps to expedite the publication of research relevant to the pandemic. If you wish, you can opt to have your paper published as soon as it is ready, rather than waiting for it to be published the scheduled Wednesday.

This means your paper will not be included in the weekly media round-up which the Society sends to journalists ahead of publication. However, it will still appear in the COVID-19 Publishing Collection which journalists will be directed to each week (<https://royalsocietypublishing.org/topic/special-collections/novel-coronavirus-outbreak>).

If you wish to have your paper considered for immediate publication, or to discuss further, please notify openscience_proofs@royalsociety.org and press@royalsociety.org when you respond to this email.

===END OF COVID-SPECIFIC TEXT -- WILL BE REMOVED AS NECESSARY BY THE EDITORIAL OFFICE===

on behalf of Kevin Padian (Subject Editor)
openscience@royalsociety.org

Reviewer comments to Author:
Reviewer: 1

Comments to the Author(s)
This ms has been revised nicely to address earlier concerns.

Appendix A

UNIVERSIDAD NACIONAL AUTÓNOMA DE MÉXICO
ESCUELA NACIONAL DE ESTUDIOS SUPERIORES
UNIDAD JURIQUILLA

September 17th, 2020

Dear Editor(s) of Royal Society Open Science,

I am pleased to resubmit for publication the revised version of MS **RSOS-200998** entitled "***Honey bee lifespan: the critical role of pre-foraging stage***".

First of all, I would like to thank the anonymous reviewer for his/her comments. The reviewer's advice has definitely improved the manuscript. We note that the reviewer did not ask for substantial revisions to experiments, analyses, or interpretation of our results. We have addressed the reviewer's comments in full, point-by-point.

Please find below answers to all the reviewer's comments.

SINCERELY,

"POR MI RAZA HABLARÁ EL ESPÍRITU"
UNAM Campus Juriquilla, Querétaro, September 17th, 2020.

ALBERTO PRADO, PhD
ASSOCIATE PROFESSOR

REVIEWER COMMENTS

This MS provides very important descriptive data on honey bee behavior, delivered with new automated monitoring technology. The data will be very useful to a variety of researchers from both applied and basic perspectives.

Recently the European Food Safety Authority (EFSA) has published a Technical Report titled "*Review of the evidence on background bee mortality*". The document has for objective to establish benchmark references of bee mortality that can be used in the risk assessment of pesticide use. Unfortunately, the EFSA document does not pay enough attention to honey bee survival data using automated monitoring systems such as RFID or optical bee counters. Our study aims precisely at providing a reference

to honey bee lifespan, in agreement with Reviewer 1, we are confident the data will be useful to scientists and policy makers.

The study is robust with excellent sample size of bees and colonies, and also exposure to different environmental conditions. The consistency of the results across these levels of analysis speaks to robust results.

We have used two different automated monitoring devices (RFID and optical counters) to record the in-and-out activity of 3786 worker honey bees. We have obtained strikingly similar results with both types of devices, as well as a strong consistency between different landscape and bee origin.

It is not clear how age at onset of foraging is calculated. The information apparently is in an R package, cited herein. It is recommended that the authors add a sentence on what assumptions were made in terms of activity changes to trigger classification as foraging flight, and ultimately as forager.

We have added a few lines explaining the details behind the *aof* function. We have also added two additional references to this part of the methodology; Gurarie et al 2009 and Requier et al. 2020. The method section now reads: "*We then calculated the age of first exit- and entrance sequence (AFE, in days), the lifespan (i.e. the age at last exit) and AOF for each bee using the aof function developed in the aof R-package (47, 54). The aof function is a simple statistical procedure, derived from the behavioural change point analysis approach (55), a well-appreciated technique of likelihood comparisons to statistically determine behavioural changes. It aims to detect a single behavioural change in univariate time series. The aof function works at the individual level thus accounting for inter-individual variation to detect, assess and quantify shifts in the temporal pattern of time-activity budgets recorded by individual life-long monitoring. Indeed, a clear breakpoint in duration, frequency and time of occurrence in the day was found between learning and foraging flights of honey bees, so-called AOF (56). Thus, AOF is estimated from three daily activities, i) the number of exit-and-entrance sequences per day, ii) the duration of these sequences, and iii) the hour of the day that they were performed. Then, these three values are averaged to give an AOF estimate in days. The aof function is freely available through the aof R-package that includes documentation, source code and examples using both simulated and empirical dataset (47, 54).*"

How long was a round trip classified as a foraging vs. orientation trip? This is an issue that investigators pay close attention to. There is general consensus, so it would be good to know whether this study uses similar metrics.

We did not classify flights/trips based on their length as other authors have done. What we classified was the stage/phase at which the worker bee was during a particular trip. This was done by first identifying a behavioural change in the worker bee's activity using the *aof* function (Requier et al. 2020). Once the AOF was determined for an individual, all of its trips prior to this moment were considered orientation flights while all the remaining trips were considered foraging flights. In the

cases where the *aof* function did not detect a change of behaviour the individuals were never considered foragers.

The MS does a good job of citing a lot of the relevant literature, but has ignored some of the earlier literature that used manual observation to arrive at some of the same conclusions, concerning the relationship between age at onset of flight and lifespan. This would include papers by Kolmes, Winston and Robinson from the 1980's.

We have added a reference to Winston and Fergusson 1985 in the introduction section. As well as Dukas and Visscher 1994 and Fukuda and Sekiguchi 1966 to the discussion section.

Indeed, some of the similarities in duration of foraging period would be very interesting to point out as bees 40 years ago did not face the same exposure to parasitic mites, pesticides, poor nutrition, and pathogens.

This is a very interesting point made by the reviewer. We have incorporated explicitly the results of Dukas and Visscher 1994, and Fukuda and Sekiguchi 1966 on honey bee foraging and life span. The discussion now reads as follows: "*Furthermore, our results on honey bee lifespan (23 to 41 days) and foraging span (8 days) are in agreement with those reported by Fukuda and Sekiguchi for spring and summer bees more than 50 years ago (lifespan between 19 and 47 days) (71) and by Dukas and Visscher (foraging span of around 7.7 days) 25 years ago (72). This is interesting since honey bees likely did not face the same anthropogenic pressures (e.g. pesticide exposure) as they do now.*"

My second suggestion is that the discussion could consider the implication of these results in light of key theories of aging and the evolution of senescence. Honey bees have figured prominently in the development of theories in this area, and I sense that the results here would make a nice contribution to some of those considerations. I understand this needs to be a short paper, so perhaps the authors can pick just a single salient aspect to comment on.

We have added the following paragraph to the discussion to mention the plasticity of honey bee senescence. The text now reads as follows: "*Honey bees have contributed to show that ageing may be a flexible process and does not necessarily result from an inevitable decline of physiological functions (65, 66). Our results on the connection between pre-foraging experience and forager lifespan further supports this view and the decoupling between mortality and chronological age (67). This represents another level of plasticity in the regulation of lifespan, besides the well-known stage-dependent trajectory of ageing (nurse vs forager) (65).*"